# AN EXPERIMENTAL STUDY OF LAYER-LEVEL TRAINING SPEED AND ITS IMPACT ON GENERALIZATION

## ABSTRACT

How optimization influences the generalization ability of a DNN is still an active area of research. This work aims to unveil and study a factor of influence: the speed at which each layer trains. In our preliminary work, we develop a visualization technique and an optimization algorithm to monitor and control the layer rotation rate, a tentative measure of layer-level training speed, and show that it has a remarkably consistent and substantial impact on generalization. Our experiments further suggest that weight decay's and adaptive gradients methods' impact on both generalization performance and speed of convergence are solely due to layer rotation rate changes compared to vanilla SGD, offering a novel interpretation of these widely used techniques, and providing supplementary evidence that layer-level training speed indeed impacts generalization. Besides these fundamental findings, we also expect that on a practical level, the tools we introduce will reduce the meta-parameter tuning required to get the best generalization out of a deep network.

## 1 INTRODUCTION

Generalization and gradient propagation are two popular themes in the deep learning literature. Concerning generalization, it has been observed that a network's ability to generalize depends on a subtle interaction between the optimization procedure and the training data (Zhang et al., 2017a; Arpit et al., 2017). Concerning gradient propagation, several works have shown that the norm of gradients can gradually increase or decrease as a function of layer depth (*i.e.* vanishing and exploding gradients (Bengio et al., 1994; Hochreiter, 1998; Glorot & Bengio, 2010)), so that some layers are trained faster than others. This work explores an interaction between generalization and the intricate nature of gradient propagation in deep networks, and focuses on the following research question: how does the speed at which each layer trains influence generalization?

Our endeavour is motivated by the following intuition: if the training data influences a neural network's generalization ability when using gradient-based optimization (Zhang et al., 2017a; Arpit et al., 2017), the input and feedback signals that a layer receives (during the network's forward and backward passes) could also influence the generalization ability induced by the layer's training. These signals result from a transformation involving the other layers of the network such that, for example, the input signals of the *last* layer could be more conducive to good generalization if the *first* layers have been significantly updated already, instead of being randomly initialized (cfr. the works on transfer learning (Donahue et al., 2014; Oquab et al., 2014)). More generally, the speed at which each layer trains during the network's training, since it directly influences how the input and feedback signals of the other layers evolve over training, could have an impact on generalization. Figure 1 supports our intuition with a toy example where training a single layer of an 11 layer MLP network, although always reaching $100\%$ train accuracy, results in different test accuracies depending on the layer's localisation in the network architecture.

Our study starts from an educated guess about how to measure layer-level training speed appropriately: we measure it through layer rotation rates, i.e. the rates at which the weight vectors of layers rotate (another approach would be to measure the norm of the weight updates at each training step, as is done in Bengio et al. (1994); Hochreiter (1998); Glorot & Bengio (2010); Pascanu et al. (2013); Arjovsky et al. (2016)). The study is then composed of the three following steps:

1. Developing tools to monitor and control layer rotation rates;

2. Using our controlling tool to systematically explore layer rotation rate configurations, varying the layers which are prioritized (first layers, last layers, or no prioritization) and the global rotation rate value (high or low rate, for all layers);[1]

3. Using our monitoring tool to study the layer rotation rates that emerge from standard training settings.[1]

The *outcomes* of our study, supported by an extensive amount of experiments, are the following:

(i) layer rotation rates have a consistent and substantial impact on generalization;

(ii) weight decay is a key ingredient for enabling the emergence of beneficial layer rotation rates during SGD training;

(iii) adaptive gradient methods' impact on generalization and training speed does not result from parameter-level, but rather from layer-level adaptation of the learning rate;

While the influence of layer-level training speed on generalization has remained unstudied, our observations thus suggest that its impact is ubiquitous in current deep learning applications. Our preliminary work offers useful guidance for meta-parameter tuning and novel insights around two widely used techniques: weight decay and adaptive gradient methods. Moreover, while layer rotation rate as a measure of layer-level training speed originated from an educated guess, the impressive consistency of its impact on generalization supports the pertinence of this choice. Our work thus also contributes to the open problem of correctly measuring layer-level training speed.

To encourage further validation of our claims, the tools and source code used to create all the figures of this paper are provided at -*github link hidden to preserve anonymity*- (code uses the Keras (Chollet et al., 2015) and TensorFlow (Agarwal et al., 2016) libraries). We also encourage interested readers to browse the supplementary material of this paper, as additional results are presented and discussed.

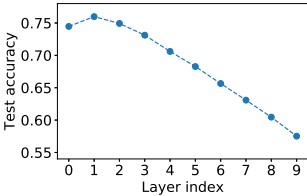

Figure 1: An eleven layer MLP network composed of 10 identical layers (each containing 784 ReLU neurons (Nair & Hinton, 2010)) is applied on a reduced MNIST dataset (LeCun et al., 1998), such that training any of the 10 layers in isolation is sufficient to get $100\%$ training accuracy. But will training of each layer result in the same test accuracy? This figure shows the test accuracy in function of the index of the trained layer (in forward pass ordering), after averaging over 10 experiments. In this specific example, the test accuracy mostly degrades with the depth of the trained layer, with a final difference of nearly $20\%$. The generalization ability induced by a layer's training is thus heavily affected by how the other layers transform the input and feedback signals it receives (everything else being equal for each layer index). Section A.1 (in Supp. Mat.) provides further discussion of this experiment.

## 2 RELATED WORK

Recent works have demonstrated that generalization in deep neural networks was largely due to the optimization procedure and its puzzling interaction with the training data (Zhang et al., 2017a; Arpit et al., 2017). Our paper discloses an aspect of the optimization procedure that influences generalization in deep learning: the rate at which each layer's weight vector is rotated. This novel

---

[1]Our preliminary study focuses on convolutional neural networks used for image classification.

factor complements batch size and global learning rate, two parameters that have been extensively studied in the light of generalization (Keskar et al., 2017; Jastrzebski et al., 2017; Smith & Le, 2017; Smith & Topin, 2017; Hoffer et al., 2017; Masters & Luschi, 2018).

The works studying the vanishing and exploding gradients problems (Bengio et al., 1994; Hochreiter, 1998; Glorot & Bengio, 2010) heavily inspired this paper. These works introduce two ideas which are central to our investigation: the notion of layer-level training speed and the fact that SGD does not necessarily train all layers at the same speed during training. Our work explores the same phenomena, but studies them in the light of generalization instead of trainability and speed of convergence.

Our paper also proposes Layca, an algorithm to control the rate at which each layer's weight is rotated during training. It is related to the works that sought solutions to the gradient propagation problems at optimization level (Pascanu et al., 2013; Hazan et al., 2015; Singh et al., 2015; Arjovsky et al., 2016; Pennington et al., 2017). These works, however, do not use weight rotation as a measure of layer-level training speed, and also focus on speed of convergence instead of generalization. Recently, a series of papers proposed optimization algorithms similar to Layca and observed an impact on generalization (Yu et al., 2017; Zhang et al., 2017b; Ginsburg et al., 2018). Section A.2 in our Supplementary Material provides evidence that these methods may be equivalent to Layca in practice, despite avoiding some of Layca's operations. Our paper thus supplements these works by providing an extensive study of the phenomena underlying their observations.

Several works have recently argued that weight decay's regularization effect emerged from its ability to increase the effective learning rate (van Laarhoven, 2017; Hoffer et al., 2018; Anonymous, 2019). A concise description of when and to what extent weight decay increases the effective learning rate is however lacking, such that using weight decay is still necessary to benefit from its regularization effect in practice.[2] Our work also analyses weight decay, but from the perspective of layer rotation rates instead of effective learning rates. We show that this new perspective enables a more succinct description of weight decay's regularizing effect, that we are able to reproduce without any additional meta-parameter tuning when using Layca, our tool for controlling layer rotation rates.

## 3    TOOLS FOR MONITORING AND CONTROLLING LAYER ROTATION RATES

This paper's goal is to study the relation between layer-level training speed and generalization. However, the notion of layer-level training speed is unclear, and its control through SGD is potentially difficult because of the intricate nature of gradient propagation (cfr. vanishing and exploding gradients). Therefore, our work starts by making an educated guess about how to measure layer-level training speed and by developing tools to monitor and control this metric during training. This section presents the metric and tools, while the impact on generalization is studied in Sections 4 and 5.

### 3.1    HOW CAN WE MEASURE LAYER-LEVEL TRAINING SPEED?

Training speed can be understood as the speed with which a model converges to its optimal solution -not to be confounded with learning rate, which is only one of the parameters that affect training speed in current deep learning applications. The notion of *layer-level* training speed is ill-posed, since a layer does not have a loss of its own: all layers optimize the same global loss function. Given a training step, how can we know by how much each layer's update contributed to the improvement of the global loss? Or, in other words, how can we measure at what rate relevant features are learned by each layer individually?

Previous work on vanishing and exploding gradients focused on the norm and variance of gradients as a measure of layer-level training speed (Bengio et al., 1994; Hochreiter, 1998; Glorot & Bengio, 2010). Provided the empirical work on activation and weight binarization during (Courbariaux & David, 2015; Rastegari et al., 2016; Hubara et al., 2016) or after training (Agrawal et al., 2014;

---

[2]van Laarhoven (2017) proposes to keep the norm of the weights fixed to 1 in order to eliminate the effect of weight decay, but not to reproduce it. Hoffer et al. (2018); Anonymous (2019) are able to reproduce the regularizing effect of weight decay by tuning the effective learning rate, but their tuning strategy consists in *copying* the effective learning rate that emerges when training the same network *with weight decay*.

---

**Algorithm 1** Layca, an algorithm that enables control over the amount of weight rotation per step for each layer through its learning rate parameter (cfr. Section 3.2).

---

**Require:** $o$, an optimizer (SGD is the default choice)
**Require:** $T$, the number of training steps
L is the number of layers in the network
**for** l=0 **to** L-1 **do**
    **Require:** $\rho_l(t)$, a layer's learning rate schedule
    **Require:** $w_0^l$, the initial multiplicative weights of layer $l$
**end for**
$t \leftarrow 0$
**while** $t < T$ **do**
    $s_t^0, ..., s_t^{L-1} = \text{getStep}(o, w_t^0, ..., w_t^{L-1})$    (get the updates of the selected optimizer)
    **for** l=0 **to** L-1 **do**
        $s_t^l \leftarrow s_t^l - \frac{(s_t^l \cdot w_t^l) w_t^l}{w_t^l \cdot w_t^l}$      (project step on space orthogonal to $w_t^l$)
        $s_t^l \leftarrow \frac{s_t^l \|w_t^l\|_2}{\|s_t^l\|_2}$        (rotation-based normalization)
        $w_{t+1}^l \leftarrow w_t^l + \rho_l(t) s_t^l$     (perform update)
        $w_{t+1}^l \leftarrow w_{t+1}^l \frac{\|w_0^l\|_2}{\|w_{t+1}^l\|_2}$    (project weights back on sphere)
    **end for**
    $t \leftarrow t + 1$
**end while**

---

Carbonnelle & De Vleeschouwer, 2018), we argue that the norm of a weight vector does not matter, but only its orientation. Therefore, we suggest to measure training speed through the rotation rate of a layer's weight vector (also denoted by layer rotation rate in this paper). More precisely, let $w_l^t$ be the flattened weight tensor of the $l^{th}$ layer at optimization step $t$, then the rotation rate of layer $l$ between steps $t_1$ and $t_2$ is defined as the angle between $w_l^{t_1}$ and $w_l^{t_2}$ divided by the number of performed steps $t_2 - t_1$. [3] In order to visualize how fast layers rotate during training, we propose to inspect how the cosine distance between each layer's current weight vector and its initialization evolves across training steps. We denote this visualization tool by *layer-wise angle deviation curves* hereafter.

## 3.2 LAYCA: AN ALGORITHM TO CONTROL LAYER ROTATION RATES

Given our definition of layer-level training speed, we now develop an algorithm to control it. Ideally, the layer rotation rates should be directly controllable with layer-wise learning rate parameters, ignoring the peculiarities of gradient propagation. We propose Layca (SGD-guided LAYer-level Controlled Amount of weight rotation), an algorithm where the layer-wise learning rates directly determine the amount of rotation performed by each layer's weight vector during an optimization step, in a direction specified by an optimizer (SGD being the default choice). Inspired by techniques for optimization on manifolds (Absil et al., 2010), and on spheres in particular, Layca applies layer-wise orthogonal projection and normalization operations on SGD's updates, as detailed in Algorithm 1. While Layca enables control over the rotation performed during one unique training step, the presence of noise or inconsistent directions can influence the overall rotation over multiple training steps in an uncontrolled way. Fortunately, such behaviour can be detected by inspecting the layer-wise angle deviation curves and did not hinder our experiments.

## 4 EXPLORATION OF LAYER ROTATION RATE CONFIGURATIONS WITH LAYCA

Section 3 provides tools to monitor and control layer rotation rates, a tentative definition of layer-level training speed. The purpose of this section is to use these tools to conduct a systematic experimental study of the relation between layer rotation rates and generalization. The experiments are

---

[3] It is worth noting that our measure focuses on weights that multiply the inputs of a layer (*e.g.* kernels of fully connected and convolutional layers). Studying and controlling the training of additive weights (biases) is left as future work.

Table 1: Summary of the tasks used for our experiments[4]

| Name | Architecture | Dataset |
|------|-------------|---------|
| C10-CNN1 | 25 layers deep CNN | CIFAR-10 |
| C100-resnet | ResNet-32 | CIFAR-100 |
| tiny-CNN | 11 layers deep CNN | Tiny ImageNet |
| C10-CNN2 | deep CNN from torch blog | CIFAR-10 + data augm. |
| C100-WRN | Wide ResNet 28-10 with 0.3 dropout | CIFAR-100 + data augm. |

conducted on three different tasks which vary in network architecture and dataset complexity, and are further described in Table 1.

### 4.1 LAYER-WISE LEARNING RATE CONFIGURATIONS

Layca enables us to specify layer rotation rate configurations by setting the layer-wise learning rates. To explore the large space of possible layer rotation rate configurations, our study restricts itself to two directions of variation. First, we vary the initial global learning rate $\rho(0)$, which affects the training speed of all the layers. During training, the global learning rate $\rho(t)$ drops following a fixed decay scheme (hence the dependence on $t$), as is common in the literature (cfr. Supplementary Material B.3). Notice that the impact of the global learning rate on generalization has already been studied when using SGD (Jastrzebski et al., 2017; Smith & Le, 2017; Smith & Topin, 2017; Hoffer et al., 2017; Masters & Luschi, 2018), but not with an algorithm like Layca where learning rate directly determines rotation rate. The second direction of variation is prioritization. We explore the impact of prioritization amongst layers by applying static, layer-wise learning rate multipliers that exponentially increase/decrease with layer depth (which is typically encountered with exploding/vanishing gradients). The multipliers are parametrized by the layer index $l$ (in forward pass ordering) and a parameter $\alpha \in [-1, 1]$ such that the learning rate of layer $l$ becomes:

$$\rho_l(t) = \begin{cases} (1-\alpha)^{5\frac{(L-1-l)}{L-1}}\rho(t) & \text{if} \quad \alpha > 0 \\ (1+\alpha)^{5\frac{l}{L-1}}\rho(t) & \text{if} \quad \alpha \leq 0 \end{cases} \tag{1}$$

Values of $\alpha$ close to $-1$ correspond to prioritizing first layers, $0$ corresponds to no prioritization, and values close to $1$ to prioritization of last layers. Visualization of the layer-wise multipliers for different $\alpha$ values is provided in Supplementary Material.

To study the impact of global learning rate, we evaluate 10 logarithmically spaced values of $\rho(0)$ $(3^{-7}, 3^{-6}, ..., 3^2)$ in the $\alpha = 0$ setting. To study the impact of prioritization, we compare 13 different values of $\alpha$, and tune the initial global learning rate $\rho(0)$ for each value separately through an iterative grid search procedure (described in Supplementary Material).

### 4.2 HOW LAYER ROTATION RATES INFLUENCE GENERALIZATION

Figure 2a shows, for each of the three tasks, the test accuracies obtained for the different $\alpha$ and $\rho(0)$ values. From these results, we extract two rules of thumb. First, the rotation rates should be uniform across layers, as prioritizing the first or last layers reduces the test accuracy by up to 20%. Second, we observe that the layer rotation rates should be selected as high as training allows it (when too high, training diverges), enabling gains of up to 30% in test accuracy. The observations generalize across the three tasks, and our preliminary exploration indicates that *layer rotation rate configurations have a consistent and substantial impact on generalization*. Let us also notice that in extreme prioritization schemes ($\mid \alpha \mid \geq 0.6$), the observations are in line with Figure 1's results, as prioritizing the first layers generalizes better than prioritizing the last layers.

Figure 2b presents the layer-wise angle deviation curves (cfr. Section 3.1) generated by the different configurations. This visualization enables us to check that, up to small deviations, Layca enables good control over the rotation rates in the tasks we consider. For example, the $\alpha = 0$ setting used

---

[4]References: ResNet (He et al., 2016), torch blog (Zagoruyko, 2015), Wide ResNet (Zagoruyko & Komodakis, 2016), CIFAR-10 (Krizhevsky & Hinton, 2009), Tiny ImageNet (Deng et al., 2009; CS231N, 2016).

in the fifth column indeed leads all layers to rotate quasi synchronously. Moreover, it is useful to visualize how the layer-wise angle deviation curves look like for the different layer rotation rate configurations, as the same visualization tool will be used in Section 5 to analyse standard training settings where layer rotation rates are not controlled during training. In particular, it is useful to remember that for the three tasks, the best generalization performance is obtained when nearly all layers synchronously reach a cosine distance of 1 from their initialization.

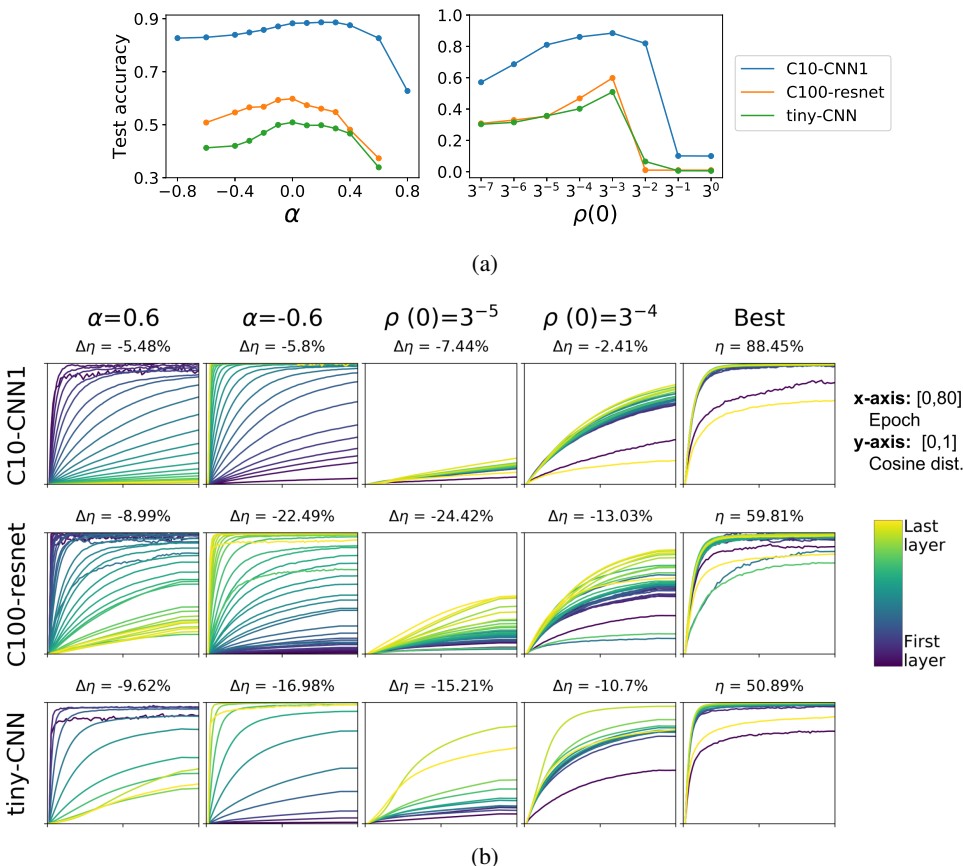

(a)

(b)

Figure 2: Analysis of the generalization ability induced by different layer rotation rate configurations (using Layca for training) on the three first tasks of Table 1. The configurations are parametrized by $\alpha$, that controls which layers are prioritized (first layers for $\alpha < 0$, last layers for $\alpha > 0$, or no prioritization for $\alpha = 0$), and $\rho(0)$, the initial global learning rate value shared by all layers. **(a)** Test accuracy in function of $\alpha$ and $\rho(0)$. Two rules of thumb emerge: layer rotation rates should be uniform across layers (*i.e.* $\alpha = 0$) and be as high as training allows it (*i.e.* high $\rho(0)$ values). **(b)** Layer-wise angle deviation curves (cfr. Section 3.1) generated by different configurations, and their accompanying test accuracy ($\eta$). $\Delta\eta$ is computed with respect to the high and uniform layer rotation rate configuration (last column), which corresponds to $\alpha = 0$ and $\rho(0) = 3^{-3}$ for the three tasks. Train accuracies are provided in Supplementary Material ($\approx 100\%$ in all configurations).

### 4.3 HOW LAYER ROTATION RATES INFLUENCE NETWORK CONVERGENCE

It is commonly assumed that vanishing and exploding gradients slow down or even prevent training of neural networks. One might thus be tempted to believe that the bad performances on the test set obtained in Figure 2 for low and/or non-uniform layer rotation rates are caused by an equally bad performance on the training set. However, not only do these layer rotation rate configurations result in close to perfect training performance (*cfr.* Supplementary Material), but they also often lead the network to converge faster than the high and uniform layer rotation rate configuration. Figure 3 depicts the loss curves obtained for different values of $\alpha$ and $\rho(0)$. It appears that the higher or

the more uniform the layer rotation rates are, the higher the plateaus in which loss curves get stuck into. The fact that plateaus are the most prominent when all layers are rotated at high rate suggests that they are caused by some kind of interference between the layers during training. Moreover, it also suggests that, following our rules of thumb, high plateaus are additional indicators of good generalization performance.

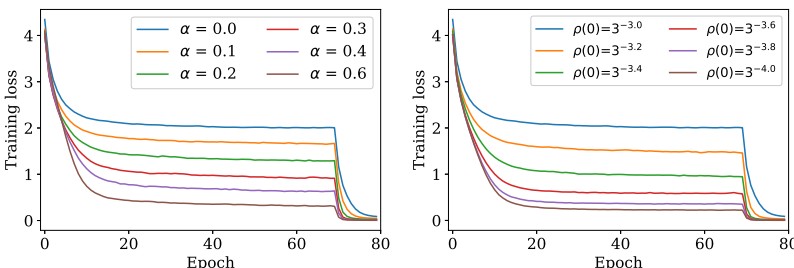

Figure 3: Loss curves obtained for different $\alpha$ and $\rho(0)$ values on the tiny-CNN task (for the two other tasks, see Supp. Mat.), using Layca for training. The more uniform or the higher the layer rotation rates, the higher the plateaus in which the loss gets stuck into. The sudden drop at epoch 70 corresponds to a reduction of the global learning rate by a factor 5.

# 5 A STUDY OF LAYER ROTATION RATES EMERGING FROM STANDARD TRAINING SETTINGS

Section 4 uses Layca to study the impact of layer rotation rates on generalization and speed of convergence in a controlled setting. This section investigates the layer rotation rates that naturally emerge when using SGD and adaptive gradient methods for training. First of all, these experiments will provide supplementary evaluation of the rules of thumb proposed in Section 4. Second, analysing SGD and adaptive gradient methods in the light of layer rotation rates' impact on generalization will provide useful insights to explain previous observations around these methods that currently escape our understanding.

The experiments of this section are performed on the three tasks used in Section 4 and on two supplementary, extensively tuned networks from state of the art (cfr. Table 1). Meta-parameters for these two tasks are taken from their original implementation when using SGD and from Wilson et al. (2017) when using adaptive gradient methods for training. For the other three tasks, the learning rate is determined by grid search over 10 logarithmically spaced values ($3^{-7}, 3^{-6}, ..., 3^2$) independently for each (task,optimizer) pair.

## 5.1 ANALYSIS OF SGD AND WEIGHT DECAY

Figure 4 ($1^{st}$ line) depicts the layer-wise angle deviation curves generated by SGD and the corresponding test accuracies for each of the five tasks. We observe that the curves are far from the ideal scenario disclosed in Section 4, where the majority of the layers synchronously reached a cosine distance of 1 from their initialization. Moreover, in accordance with our rules of thumb, SGD induces a considerably lower test performance than Layca (in the high and uniform rotation rate configuration). Extensive tuning of the learning rate did not help SGD to solve its two systematic problems: 1) layers don't train at the same speed and 2) the layers' weights stop rotating before reaching a cosine distance of 1.

At this point, it is tempting to believe that Layca will improve the performance of all the deep networks trained with SGD. However, we observed that, with the unexpected help of weight decay (*i.e.* $L_2$-regularization), SGD gains the ability to induce high and uniform layer rotation rates and the accompanying good test performance. Figure 4 ($2^{nd}$ line) displays, for the 5 tasks, the layer-wise angle deviation curves generated by SGD when combined with weight decay. We observe that all layers are rotated synchronously, reaching a cosine distance of 1 from their initialization, and that the resulting test performances are on par with the ones obtained with Layca. This experiment not

only provides important supplementary evidence for our rules of thumb, but also suggests a radically novel explanation of weight decay's regularization ability in deep nets: ***weight decay enables the emergence of high and uniform layer rotation rates during SGD training***.[5] On a practical level, since the same regularization effect can be elegantly achieved with tools that control layer rotation rates, without an extra parameter to tune, our results could potentially lead weight decay to disappear from the standard deep learning toolkit.

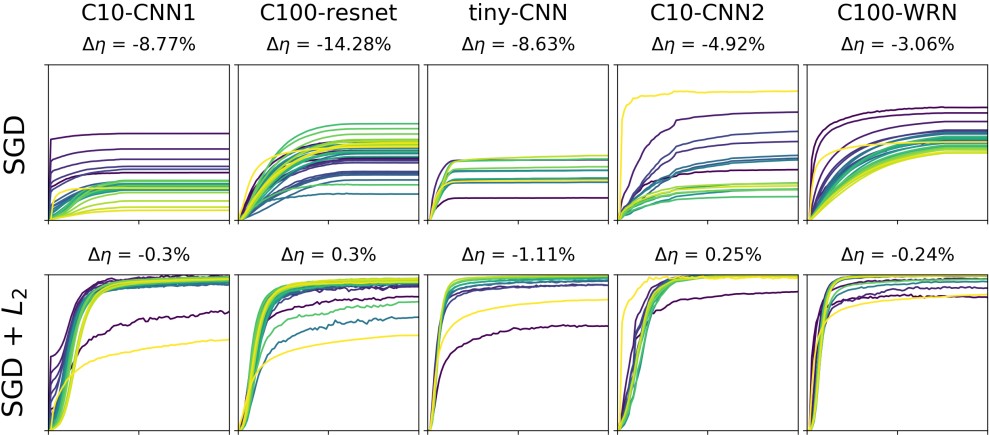

Figure 4: Layer-wise angle deviation curves and the corresponding test accuracies generated by SGD without ($1^{st}$ line) or with ($2^{nd}$ line) weight decay. Colour code, axes and $\Delta\eta$ computation are the same as in Figure 2b.[6]Despite extensive learning rate tuning, SGD without weight decay induces test performances that are significantly below Layca. These results are coherent with our rules of thumb, as SGD is not able to induce high and uniform layer rotation rates (cfr. $5^{th}$ column of Figure 2b). Surprisingly, weight decay solves SGD's problems, leading to high and uniform layer rotation rates and test accuracies that are on par with Layca.

## 5.2 ANALYSIS OF ADAPTIVE GRADIENT METHODS

The recent years have seen the rise of adaptive gradient methods in the context of machine learning (*e.g.* RMSProp (Tieleman & Hinton, 2012), Adagrad (Duchi et al., 2011), Adam (Kingma & Ba, 2015)). Initially introduced for improving training speed, Wilson et al. (2017) observed that these methods also had a considerable impact on generalization. Figure 5 provides the layer-wise angle deviation curves and test accuracies obtained when using adaptive gradient methods for training of the 5 tasks described in Table 1. Again, our rules of thumb can be applied: the overall worse generalization ability compared to Layca corresponds to low and/or non-uniform layer rotation rate configurations.

We also observe that the layer rotation rates of adaptive gradient methods are considerably different from the ones induced by SGD (cfr. Figure 4). For example, adaptive gradient methods tend to prioritize the last layers while SGD usually prioritizes the first layers in the tasks we study. Previous results of this paper indicate that these differences could be the reason behind adaptive gradient methods' influence on both training speed and generalization. Figure 6 confirms this hypothesis by showing that the training and test curves of adaptive gradient methods become indistinguishable from their non-adaptive equivalents when Layca is used to enforce a fixed layer rotation rate configuration. Additional evidence is provided in Supplementary Material (Section A.4), where we observe experimentally that the learning rates of adaptive gradient methods vary mostly across layers and negligibly inside layers.

---

[5]Notice that this observation is also consistent with the systematic occurrence of high plateaus (*cfr.* Section 4.3) in the loss curves of state of the art networks (He et al., 2016; Zagoruyko & Komodakis, 2016) (which usually use SGD with weight decay).

[6]On the two supplementary tasks, the reference accuracy ($\eta$) is also obtained by Layca with $\alpha = 0$ and $\rho(0) = 3^{-3}$ and equals 92.33% and 80.69% on C10-CNN2 and C100-WRN respectively.

Adaptive gradient methods were initially introduced with a focus on parameter-level adaptation of the learning rate in order to improve optimization in the presence of sparse gradients or optimization in on-line and non-stationary settings (Kingma & Ba, 2015). Our work offers and empirically validates a radically different understanding of adaptive gradient methods' role in deep learning, that solely relies on their impact on the rate at which each layer's weights change during training. This new perspective reveals that *adaptive gradient methods' parameter-level adaptation of the learning rate is not necessary (layer-level adaptation suffices)*, and that their impact on DNN training will probably be better explained by further study of layer-level training speed rather than the sparsity of gradients or the on-line aspect of the optimization procedure.

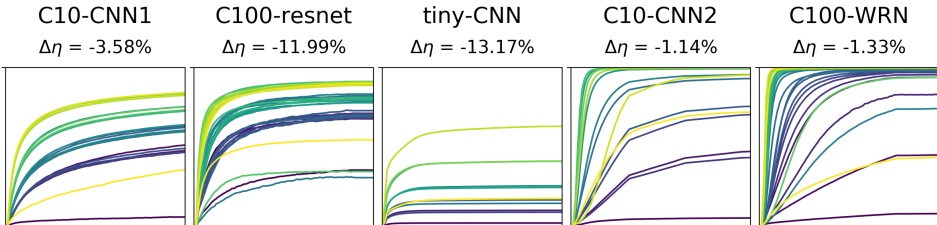

Figure 5: Layer-wise angle deviation curves and the corresponding test accuracies generated by adaptive gradient methods (RMSProp, Adam, Adagrad, RMSProp+$L_2$ and Adam+$L_2$ respectively for each task/column) after extensive learning rate tuning. Colour code, axes and $\Delta\eta$ computation are the same as in Figure 2b. Our rules of thumb still apply: the overall worse generalization ability compared to Layca corresponds to low and/or non-uniform layer rotation rate configurations. We also notice that the curves are significantly different from the ones of SGD (Figure 4). For example, on the C10-CNN1 task, the last layers train faster than the first ones when RMSProp is used for training while the opposite happens when SGD is used. Could this explain adaptive gradient methods' impact on training speed and generalization?

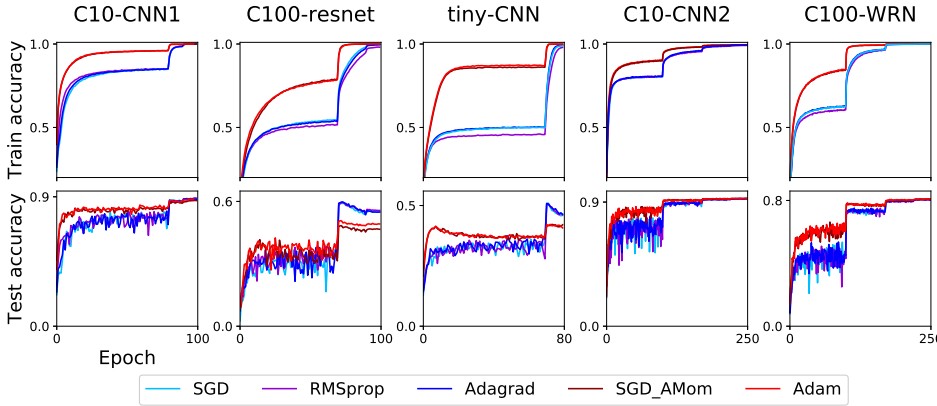

Figure 6: Training and test curves of adaptive gradient methods and their non-adaptive equivalents[7] when Layca is applied on the updates of each method to fix the layer rotation rate configuration. The curves become indistinguishable, suggesting that adaptive gradient methods' impact on training speed and generalization is only due to their influence on layer rotation rates, and thus to their layer-level adaptation of the learning rate.

## 6 EMPHASIZING THE REMARKABLE CONSISTENCY OF LAYER ROTATION RATES' IMPACT ON GENERALIZATION

In order to emphasize the remarkable consistency by which layer rotation rates influence generalization, this section provides evidence that the norm of updates, another sensible metric of layer-level

---

[7]SGD_AMom corresponds to SGD with a momentum scheme similar to Adam (see Supp. Mat.).

training speed (Bengio et al., 1994; Hochreiter, 1998; Glorot & Bengio, 2010; Pascanu et al., 2013; Arjovsky et al., 2016), does not demonstrate this consistency at all. This result supports the idea that layer rotation rates are related to a more fundamental aspect of learning, and probably constitute a relevant metric to study layer-level training speed.

The main difference between layer rotation rates and the norm of updates as layer-level training speed metrics, is that the latter doesn't take the norm of the current layer's weights into account. Our experiment targets this difference specifically, by testing if the relation between the studied metric and generalization remains consistent when the initial weights of all layers are rescaled by a constant factor $f$. The independence of layer-level metrics to rescaling of the weights is especially important as, with typical weight initialization schemes, weights of different layers have different scales (e.g. Glorot & Bengio (2010)).

Using Layca to control layer rotation rates, and block-normalized SGD (Yu et al., 2017) to control the norm of weight updates, we study for several factors $f$ the relation between test accuracy and the initial global learning rate in Figure 7 (experiment is conducted on a lighter version of the C10-CNN1 task). The difference is striking. In contrast with block-normalized SGD, Layca remains very consistent across scaling factors. For example, the optimal learning rate is $3^{-3}$ in all cases. On the contrary, when block-normalized SGD is used, the curve that relates learning rates to train or test accuracy shifts every time the scaling factor is increased, and so does the optimal learning rate.

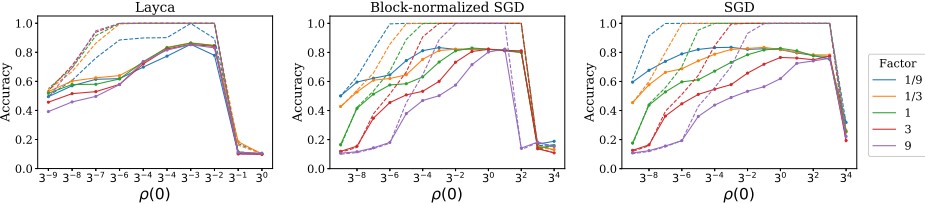

Figure 7: Consistency comparison of layer rotation rates and the norm of updates (another sensible metric of layer-level training speed). Using Layca (left) or block-normalized SGD (center) to control both metrics, we test if the relation between global initial learning rates and train/test accuracy remains consistent when the initial weights of all layers are rescaled by a constant factor $f$. The difference is striking: block-normalized SGD does not demonstrate Layca's consistency at all. SGD is also added for comparison (right). Dotted lines correspond to train accuracy, full lines to test accuracy.

# 7 CONCLUSION

Inspired by the works on generalization and gradient propagation in deep networks, this paper's ambition is to disclose and study a novel way, unique to deep learning, by which optimization influences generalization: through the speed at which each layer trains. While the premises of our work are based on intuitions that escape any theoretical framework, the value of our tools and claims is supported by the consistent experimental results and the useful insights they provide. Indeed, the rules of thumb we extract about the relation between layer rotation rates and generalization could be successfully applied to all the considered tasks and training settings, explaining substantial differences in test accuracies (sometimes of the order of $20\%$). Moreover, we show considerable evidence that this novel way by which optimization impacts generalization could be the reason behind weight decay's and adaptive gradient methods' generalization properties, which have remained a mystery despite their ubiquity in current deep learning applications. Both methods, which are source of heavy meta-parameter tuning, could even become obsolete as practitioners start using our tools, exemplifying the practical benefit of our work.

ACKNOWLEDGEMENTS

To be filled in

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

## SUPPLEMENTARY MATERIAL

The supplementary material of this paper is divided into two sections. Section A contains supplementary results, which are not essential for the main message of the paper but could be useful for researchers interested in pursuing our line of work. Section B contains supplementary information about the experimental procedure used for generating the results of our paper.

## A   SUPPLEMENTARY RESULTS

### A.1   DISCUSSING THE FIRST LAYERS' SUPERIOR PERFORMANCE IN FIGURE 1.

#### A.1.1   STUDYING THE IMPACT OF THE INITIALIZATION SCHEME

Figure 1 shows that in our toy example, the first layers of the MLP network receive input and feedback signals of better quality, i.e. that lead to better generalization properties (hereafter, *quality* of signals will refer to the generalization performance that results from training the layers that receive these signals, high quality signals leading to good generalization performance). Importantly, this example shows that the quality of the original input and error signals is altered when going through the layers during the forward and backward pass. It is interesting to compare how different initialization schemes alter the quality of the input and feedback signals. In particular, orthogonal initializations' ability to faithfully propagate signals through layers has been praised in several papers (Saxe et al., 2014; Xie et al., 2017; Pennington et al., 2017). Figure 8 reproduces the experiment with an orthogonal initialization scheme. We observe that all layers reach higher test accuracy, and that the gap between the first and last layers decreases. This suggests that indeed, input and feedback signals are less altered when going through layers with orthogonal weight matrices.

#### A.1.2   IMPROVING THEIR OWN FEEDBACK: THE FIRST LAYERS' SECRET TRICK?

To explain the first layers' superior performance, the first idea that came to our mind was that, in the randomly initialized network, the quality of layer inputs degrades faster during the forward pass than the quality of feedbacks does during the backward pass. Accordingly, the first layers gained their good test accuracy from their good quality input signals (since these do not result from many random transformations applied on the original inputs) and good feedback signals (since these are robust to random transformations).

This interpretation comes from a static view of the phenomenon: it implicitly assumes that the transformations applied to the input and feedback signals before reaching a layer don't change when this layer is trained in isolation. However, while the transformations applied on the input signals of a layer are not modified, the transformations applied on the feedback signals change in a non-negligible way: a layer's training influences the way errors backpropagate through every subsequent layer, because ReLU's derivative depends on the activations of the forward pass. Thus, the backwards pass, which transforms the feedback signal before reaching the trained layer, is not a series of random non-linear transformations any more after training has started.

We believe that this could be a key factor that enables the first layers' superior performance. Indeed, in addition to their good quality input signals, the first layers could also potentially receive good quality feedback signals, as these can be improved by the layers' training. Figure 8 provides some evidence in favour of this hypothesis. Using the Silhouette coefficient (Kaufman & Rousseeuw, 2009) (with cosine distance as distance metric), the figure shows that even when only the first layer is trained, the feedback it receives gets more correlated with the classes/targets through training, which we believe could be a sign of better signal quality (it remains to be proven however).

### A.2   ALL OPERATIONS OF LAYCA ARE NOT ALWAYS NECESSARY IN PRACTICE.

The 4 main operations of Layca are repeated in Algorithm 2. The first operation projects the step on the space orthogonal to the current weights of the layer. Having a step orthogonal to the current weights is necessary for operation 2 to normalize the rotation performed during the update. However, since a layer typically has more than thousands of parameters (*i.e.* has a lot of dimensions), the step proposed by an optimizer has a high probability of being approximately orthogonal to the current

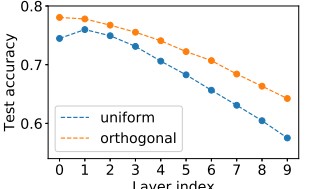 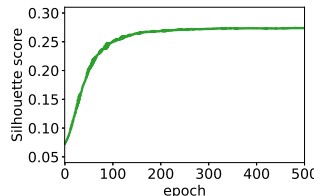

Figure 8: **left**: Studying the impact of initialization on the MNIST experiment of Figure 1. We observe that orthogonal initialization, known for its faithful propagation of signals, increases the test accuracy of every layer and slightly reduces the generalization gap between the first and last layers. **right**: Supplementary result suggesting that the capacity of layers to improve their own feedback signal could be a key asset enabling the first layers' superior generalization performance in Figure 1. Indeed, this figure shows how the Silhouette score of the first layer's feedback with respect to the classes/targets increases even when only the first layer is trained.

---

**Algorithm 2** Main operations of Layca (cfr. Algorithm 1). We've noticed that in practice, operations 1 and 4 are not strictly necessary for controlling layer rotation rates.

---

$s_t^0, ..., s_t^{L-1} = \text{getStep}(o, w_t^0, ..., w_t^{L-1})$    (get the updates of the selected optimizer)

**for** l=0 **to** L-1 **do**

$\quad s_t^l \leftarrow s_t^l - \frac{(s_t^l \cdot w_t^l) w_t^l}{w_t^l \cdot w_t^l}$        (1: project step on space orthogonal to $w_t^l$)

$\quad s_t^l \leftarrow \frac{s_t^l \|w_t^l\|_2}{\|s_t^l\|_2}$        (2: rotation-based normalization)

$\quad w_{t+1}^l \leftarrow w_t^l + \rho_l(t) s_t^l$        (3: perform update)

$\quad w_{t+1}^l \leftarrow w_{t+1}^l \frac{\|w_0^l\|_2}{\|w_{t+1}^l\|_2}$        (4: project weights back on sphere)

**end for**

---

weights. Explicitly orthogonalizing the step and the weights through operation 1 is thus potentially redundant.

Operation 4 keeps the norm of weights fixed during the whole training process. First, this operation emphasizes our claim that the norm of weights doesn't really matter. Indeed, disabling changes to the norm of weights doesn't prevent the network from reaching 100% training accuracy. Second, this operation prevents the weights from increasing too much (the first three operations lead the norm of weights to increase at every training step), which causes numerical problems. However, this operation is not fundamental for controlling the layer rotation rates.

We experimented with a sub-version of Layca that does not perform Layca's operations 1 and 4. Interestingly, the resulting algorithm is equivalent to $\text{NG}_{\text{adap}}$ and LARS introduced by Yu et al. (2017) and Ginsburg et al. (2018) respectively. Both works reported improved test performance when using this algorithm. Figure 9 shows the layer-wise angle deviation curves and associated test accuracies when applying LARS (or equivalently, $\text{NG}_{\text{adap}}$) on tasks C10-CNN1, C100-resnet and tiny-CNN.[8] The layer rotation rate configuration parameters are $\alpha = 0$ and $\rho(0) = 3^{-3}$. We observe that this configuration also induces high and uniform rotation rates, and that the test accuracies are on par with Layca. This observation indicates that operations 1 and 4 of Layca can be removed in at least some practical applications.

### A.3 IMPACT OF LAYER ROTATION RATES ON CONVERGENCE FOR C10-CNN1 AND C100-RESNET TASKS.

Figure 3 shows on the tiny-CNN task that the $\alpha$ and $\rho(0)$ parameters, which determine the layer rotation rate configuration when using Layca for training, enable precise control over the height of

---

[8]While the norm of each layer's weight vector was not fixed by LARS, we still had to limit the amount of norm increase per training step to prevent numerical errors. We limited it to 0.0001 times the initial norm of each layer's weight vector.

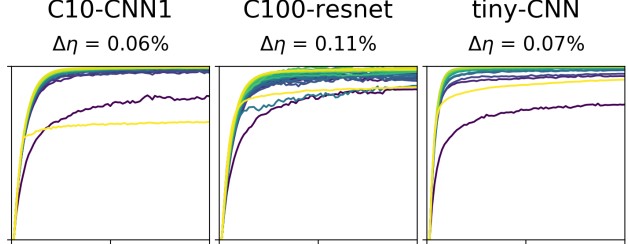

Figure 9: Layer-wise angle deviation curves and the corresponding test accuracies generated by LARS with $\alpha = 0$ and $\rho(0) = 3^{-3}$. Colour code, axes and $\Delta\eta$ computation are the same as in Figure 2b. Although not performing operations 1 and 4 of Algorithm 2, LARS seems to control layer rotation rates as well as Layca. Indeed, the layer-wise angle deviation curves are indistinguishable from the ones in the $5^{th}$ column of Figure 2b, and the test accuracies are nearly identical.

the plateaus in which the loss curve gets stuck into. Figure 10 extends the results to the C10-CNN1 and C100-resnet tasks. Conclusions are identical. Moreover, the experiment on C10-CNN1 was performed with negative $\alpha$ values, showing that prioritizing the training of the first layers of the network also decreases the height of the plateaus (at the cost of generalization ability however).

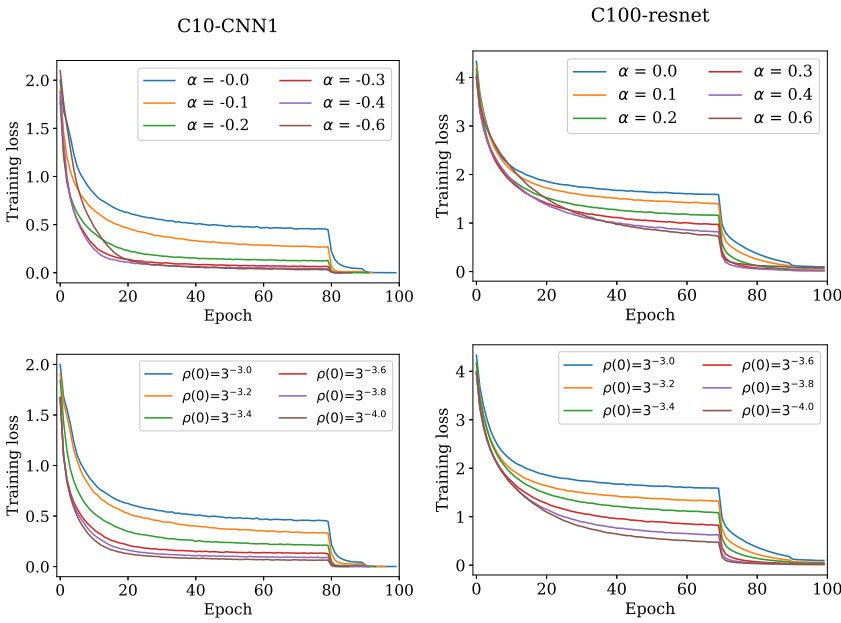

Figure 10: Loss curves of C10-CNN1 and C100-resnet for different $\alpha$ and $\rho(0)$ values. The results confirm the observations made on tiny-CNN, and extend the analysis to negative $\alpha$ values.

## A.4 FURTHER ANALYSIS OF ADAPTIVE GRADIENT METHODS.

Section 5.2 shows considerable evidence that adaptive gradient methods' impact on training speed and generalization is solely due to their influence on layer rotation rates. The key element distinguishing adaptive gradient methods from their non-adaptative equivalents is a parameter-level tuning of the learning rate based on the statistics of each parameter's partial derivative. Our results suggest that the resulting parameter-level learning rates differ mostly across layers and negligibly inside layers. To test this claim, we monitored Adam's estimate of the second raw moment when training on the C10-CNN1 task. The estimate is computed by:

$$v_t = \beta_2 \cdot v_{t-1} + (1 - \beta_2) \cdot g_t^2$$

where $g_t$ and $v_t$ are vectors containing respectively the gradient and the estimates of the second raw moment at training step $t$, and $\beta_2$ is a parameter specifying the decay rate of the moment estimate. While our experiment focuses on Adam, the other adaptive methods (RMSprop, Adagrad) also use statistics of the squared gradients to compute parameter-level learning rates.

Figure 11 displays the $10^{th}$, $50^{th}$ and $90^{th}$ percentiles of the moment estimations, for each layer separately, as measured at the end of epochs 1, 10 and 50. The conclusion is clear: the estimates indeed vary much more across layers than inside layers. While parameter-level adaptivity could make sense in other applications, deep neural networks seem to lend themselves better to layer-level adaptivity.

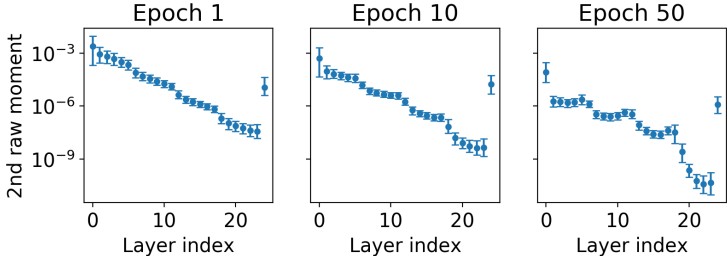

Figure 11: Adam's parameter-wise estimates of the second raw moment (uncentered variance) of the gradient during training on C10-CNN1, described for each layer separately through the $10^{th}$, $50^{th}$ and $90^{th}$ percentiles (represented by the lower bar, the bullet point, and the upper bar respectively for each layer index). The results provide supplementary evidence that the parameter-level statistics used by adaptive gradient methods vary mostly between layers and negligibly inside layers.

# B SUPPLEMENTARY INFORMATION

## B.1 VISUALIZING THE $\alpha$ PARAMETER.

The $\alpha$ parameter is used in Section 4 to characterize the layer prioritization schemes used during training. While the specific parametrization is provided in Equation 1, Figure 12 provides a graphical illustration of it.

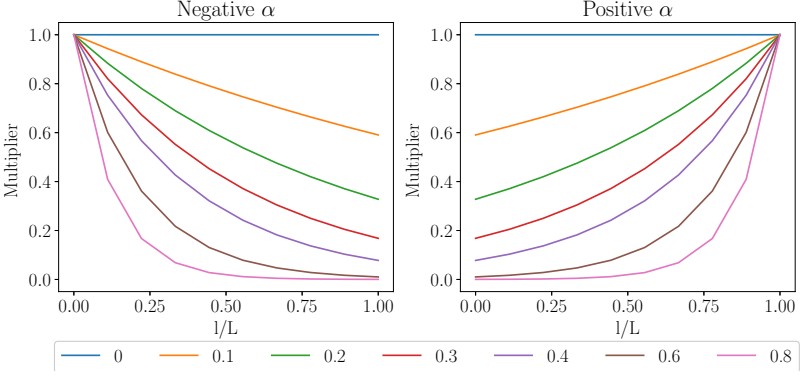

Figure 12: Visualization of the prioritization schemes as parametrized by $\alpha$ (cfr. Section 4). The colours of the lines represent the absolute value of $\alpha$. Illustration is separated for prioritization of the first layers (negative $\alpha$ values) and of the last layers (positive $\alpha$ values). The layer-wise learning rate multipliers (y-axis) depend on the layer's location in the network (x-axis), which is represented by the layer index $l$ (in forward pass ordering) divided by the number of layers $L$.

## B.2 GRID SEARCH PROCEDURE FOR LEARNING RATE SELECTION IN SECTION 4

In Section 4, the global initial learning rate parameter $\rho(0)$ is optimized through grid search for each $\alpha$ value. While in the other experiments, learning rate selection is performed through grid search over 10 logarithmically spaced values $(3^{-7}, 3^{-6}, ..., 3^2)$, such method is to demanding computationally for this experiment (it must be repeated 39 times = 13 $\alpha$ values * 3 tasks).

The grid search procedure starts by trying 3 values: $3^{-4}, 3^{-3}$ and $3^{-2}$. Then, iteratively until convergence, if the current best value is the lowest or highest of the tried values, the next value (lower or higher by a factor 3 respectively) is tried. After this first stage, the optimal $\rho(0)$ values of two successive $\alpha$ values are compared (successive after sorting the values in increasing order). If the optima of two successive $\alpha$ values are different by a factor 3, the intermediate $\rho(0)$ is also tried (current value multiplied or divided by $3^{0.5}$). The increased precision of the second stage was used to get smoother curves in Figure 2a.

## B.3 LEARNING RATE DECAY SCHEMES

Our work uses standard learning rate decay schemes, as follows:

- C10-CNN1: 100 epochs and a reduction of the learning rate by a factor 5 at epochs 80, 90 and 97

- C100-resnet: 100 epochs and a reduction of the learning rate by a factor 10 at epochs 70, 90 and 97

- tiny-CNN: 80 epochs and a reduction of the learning rate by a factor 5 at epoch 70

## B.4 TRAINING ERRORS ASSOCIATED TO THE LAYER-WISE ANGLE DEVIATION CURVES.

In Figures 2b, 4 and 5, the test accuracies corresponding to each visualization of the layer-wise angle deviation curves are provided. While it is briefly mentioned that training accuracy is close to perfect in most cases (cfr. Section 4.3), Tables 2, 3 and 4 provide the exact values for completeness.

Table 2: Train accuracies associated to Figure 2b

|  | $\alpha = 0.6$ | $\alpha = -0.6$ | $\rho(0) = 3^{-5}$ | $\rho(0) = 3^{-4}$ | Best |
|---|---|---|---|---|---|
| C10-CNN1 | 100% | 99.55% | 100% | 100% | 99.99% |
| C100-resnet | 97.38% | 97.9% | 99.87% | 99.99% | 99.75% |
| tiny-CNN | 99.98% | 98.64% | 99.97% | 99.97% | 98.91% |

Table 3: Train accuracies associated to Figure 4

|  | C10-CNN1 | C100-resnet | tiny-CNN | C10-CNN2 | C100-WRN |
|---|---|---|---|---|---|
| SGD | 100% | 100% | 100% | 91% | 100% |
| SGD + $L_2$ | 100% | 100% | 89.8% | 99.5% | 100% |

Table 4: Train accuracies associated to Figure 5

|  | C10-CNN1 | C100-resnet | tiny-CNN | C10-CNN2 | C100-WRN |
|---|---|---|---|---|---|
| Adaptive methods | 100% | 99.98% | 99.97% | 98.72% | 99.92% |

### B.5 MOMENTUM SCHEME USED BY SGD_AMOM AND ADAM.

SGD_AMom was designed for Section 5.2, as a non-adaptive equivalent of Adam. In particular, SGD_AMom uses the same momentum scheme as Adam:

$$
\begin{aligned}
v_t &= m \cdot v_{t-1} + (1 - m) \cdot g_t \\
w_t &= w_{t-1} - \rho \cdot v_t
\end{aligned}
$$

where $g_t$ is the gradient at step $t$, $\rho$ the learning rate, $m$ the momentum parameter.

### B.6 FURTHER INFORMATION.

Other details about our experiments can be found in our code at *-github link hidden to preserve anonymity-*.

