# OpenReview forum: "An experimental study of layer-level training speed and its impact on generalization"
_ICLR.cc/2019/Conference_

### Official Review · AnonReviewer1 · 2018-10-19
**Impressive theme and motivation, but limited contribution**

**Rating:** 5
**Confidence:** 2

**Review:**

This paper insists layer-level training speed is crucial for generalization ability. The layer-level training speed is measured by  angle between weights at different time stamps in this paper. To control the amount of weight rotation, which means the degree of angle movement, this paper proposes a new algorithm, Layca. This algorithm projects the gradient vector of SGD (or update vector of other variants) onto the space orthogonal to the current weight vector, and adjust the length of the update vector to achieve the desirable angle movement. This paper conducted several experiments to verify the helpfulness of Layca.

This paper have an impressive theme, the layer-level training speed is important to have a strong generalization power for CNNs. To verify this hypothesis, this paper proposes a simply SGD-variant to control the amount of weight rotation for showing its impact on generalization. This experimental study shows many insights about how the amount of weight rotation affect the generalization power of CNN family. However, the contribution of this paper is limited. I thought this paper lacks the discussion of how much the layer-level training speed is important. This paper shows the Figure 1 as one clue, but this figure shows the importance of each layer for generalization, not the importance of the layer-level training speed. It is better to show how and how much it is important to consider the layer-level training speed carefully, especially compared with the current state-of-the-art CNN optimization methods or plain SGD (like performance difference).

In addition, figures shown in this paper are quite hard to read. Too many figures, too many lines, no legends, and these lines are heavily overlapped. If this paper is accepted and will be published, I strongly recommend authors choose some important figures and lines to make these visible, and move others to supplementary material.

---

> ### Author Response · Authors · 2018-11-28
> **Detailed answer to the comments of the reviewer**
>
> We are happy that the reviewer appreciates the research question introduced by our paper. The bulk of our rebuttal is contained in the two comments addressed to all reviewers. We add here answers to your specific questions.
>
> If we understand the comment well, the reviewer believes our contribution is limited because we study layer rotation rates and the training of layers in isolation, instead of layer-level training speed. As can be deduced from the other reviews and the bulk of our rebuttal, studying layer-level training speed is a difficult task because there is no clear way to measure it. As long as no exact measure has been found, studying layer-level training speed directly is simply impossible. In our paper, we thus make the choice of measuring it indirectly/approximately, by training layers in isolation, or by monitoring layer rotation rates.
>
> We believe this approach provides important contributions to the community. First, although indirectly, our provides substantial evidence that layer-level training speed’s impact on generalization is ubiquitous in current deep learning applications, and thus that its study should be taken seriously (to our knowledge, no paper has been published on this research question yet). Second, we provide evidence that layer rotation rates are a pertinent measure of layer-level training speed, as they have a remarkably consistent relation to generalization. Third, even if we completely ignore our contributions around layer-level training speed, our paper still provides useful guidelines for meta-parameter tuning and fundamental insights around weight decay and adaptive gradient methods.
>
> We hope our rebuttal clarifies our methodology, and highlights the significance of our contributions. We thank the reviewer again for his efforts for reviewing our paper.

---

### Official Review · AnonReviewer2 · 2018-11-02
**There are some interesting ideas but I am not yet convinced that the measured quantity is really the one we should care about**

**Rating:** 5
**Confidence:** 4

**Review:**

Paper summary: The authors propose layer rotation speed as a measure of layer-wise training speed and introduce the Layca optimizer as a means of enforcing uniform layer rotation speed throughout the network. They show empirically that layer rotation speed is linked to the generalization performance of deep neural networks and that weight decay induces uniform layer rotation speeds.


Detailed comments:

Overall, I felt that the paper introduced some interesting ideas but I was not left convinced that layer rotation speed is the correct measure of layer training speed. I hope that the authors can clarify this based on my questions and comments below.

1) In the introduction you refer to input and feedback signals to a layer, I assume this refers to the forward and backward pass. As I understand it, this intuition and the findings of Figure 1 do not immediately relate to the notion of layer rotation speed during training. Could you clarify what you mean by "the input and feedback signals that a layer receives could also influence the generalization ability induced by the layer's training", to me this statement seems obvious as input+feedback signals contains training entirely.

2) In Figure 1 you show that when training one chosen layer and keeping the others fixed, if the chosen layer is deeper into the network the test accuracy is worse. I wonder to what extent this might be remedied by initialization. For example, one might expect that when sampling random square matrices there are some very small eigenvalues which "kill" information in the forward pass. If we train a layer deep in the network it may have access to less information from the data than one earlier on. Whereas training an earlier layer could allow this layer to shift mass into the parts of the eigenspace which are well represented (so-to-speak) in the future layers. Have you thought about this at all? One simple way to evaluate this would be to initialize the weights to be random orthogonal matrices, ensuring that the eigenvalues are equal. With that said, I thought that this was an interesting experiment with a fairly surprising outcome!

3) In related work you discuss the vanishing and exploding gradients problems in terms of layer-level training speed. I think that another relevant research direction may be dynamical isometry [1] which solves this problem by restricting all singular values of the Jacobian matrix to be close to 1. These ideas may also be relevant when discussing Layca and layer-rotation.

4) I found section 3.1. a little unconvincing. It is not obvious to me that layer rotation speed is necessarily a good measure of training speed. In fact, there are many updates which have large cosine distance (as you define it) but do not change the network function (for example, permuting the weight matrices in fully-connected networks). Why is the rotation defined through a vectorization of the weight matrix as opposed to e.g. the polar decomposition? Is this a computational issue? Similarly, in section 3.2 you liken Layca to optimization on a manifold but I am not convinced that this makes sense for matrices which inherently have some structure (e.g, perhaps the Stiefel manifold would be more meaningful).

6) Figure 2 shows that uniform rotation leads to improved test accuracy. But could it be the case that controlling the effective learning rate is sufficient (and layer rotation is one way to achieve this)? For example, we might take the sign of the update and use this to ensure that each weight matrix has the same effective learning rate (something like [2]). Do you expect this would have a similar effect? If not, what is unique about layer rotation that provides good test accuracy?

7) You claim that SGD and adaptive methods with weight decay works without taking extra care to control the layer-rotation rate, as weight decay provides a similar effect. Firstly, you use weight decay and L2 regularization interchangeable, could you be explicit about exactly which you mean (see e.g. [3]). Assuming you mean weight decay (and not L2 regularization), then this could also be due to the effective learning rate ([4,5,6]) which may have some interaction with layer rotation rate (i.e. Figure 4). In summary, I would have liked to see an explanation for why weight decay leads to uniform rotation speeds.

8) If I understand correctly, Figure 5 shows 5 tasks and reportedly 5 optimization schemes - each on a different task? It seems more reasonable to compare these on the same task.

Overall I felt that the paper had some interesting contributions and a fairly comprehensive empirical study. However, I do not feel that the paper gives adequate attention to the notion of effective learning rate induced by weight decay and I was not totally convinced that the way layer rotation speed is defined is the correct way.

Minor comments:

- A lot of white space and a large caption for Figure 1.
- Section 4 opens with "monitor and control", but I think the latter is really presented in section 4 and not section 3.
- I think a diagram of the projection step of the Layca algorithm would be informative (for 2D weight vector).
- Why does `5` appear in equation 1? Is this an arbitrary choice?
- Some of the lighter colors in e.g. Figure 2(b) made some lines hard to read when printed. I do not believe that this affected the image significantly.

Clarity: The paper is well written and is easy to understand. Some of the figures in the experiments are a little cluttered and the lighter colors can be hard to see (e.g. Fig 2(b)), but this is minor.

Significance: The paper presents an interesting view point but I am not convinced that it offers as strong an explanation for these phenomena as other approaches. I believe with some more clarification the results could become more significant. My review score hinges mostly on the interaction between layer rotation speed and the effective layer-wise learning rate.

Originality: To my knowledge, the ideas are presented in the paper are original. In particular, this is a novel way to characterize layer-wise training speed.


References:

[1] Pennington et al. "Resurrecting the sigmoid in deep learning through dynamical isometry: theory and practice" https://arxiv.org/abs/1711.04735
[2] Bernstein et al. "signSGD: Compressed Optimisation for Non-Convex Problems" https://arxiv.org/abs/1802.04434
[3] Loshchilov et al. "Fixing Weight Decay Regularization in Adam", https://arxiv.org/pdf/1711.05101.pdf
[4] Laarhoven, "L2 Regularization versus Batch and Weight Normalization" https://arxiv.org/abs/1706.05350
[5] Hoffer et al. "Norm matters: efficient and accurate normalization schemes in deep networks" https://arxiv.org/abs/1803.01814
[6] Anonymous, "Three Mechanisms of Weight Decay Regularization" https://openreview.net/forum?id=B1lz-3Rct7   (Another ICLR 2019 submission)

---

> ### Author Response · Authors · 2018-11-28
> **Detailed answer to the comments of the reviewer (2/2)**
>
> R7) Please note that our analysis of weight decay is only performed on SGD, and not on adaptive gradient methods (cfr. general comment to all reviewers that clarifies our analysis of adaptive gradient methods). In the case of SGD, weight decay and L2 regularization are equivalent, which explains why we use both interchangeably.
>
> As mentioned by the reviewer, several works have recently argued that weight decay's regularization effect emerged from its ability to increase the effective learning rate. These works, however, do not provide a concise description of when and to what extent weight decay changes the effective learning rate, such that using weight decay is still necessary to benefit from its regularization effect in practice. Our work also analyses weight decay, but from the perspective of layer rotation rates instead of effective learning rates. These two perspectives differ in two important ways. First, learning rates are not monitored on a per layer basis, while layer rotation rates are. Second, learning rates are independent of the gradient signal, while layer rotation rates measure weight updates, which depend on the learning rate and the gradient signal. We show that our new perspective enables a more succinct description of weight decay's regularizing effect, that we are able to reproduce without any additional meta-parameter tuning when using Layca, our tool for controlling layer rotation rates. This discussion has been added to the related work section. We thank the reviewer for highlighting this pertinent line of work.
>
> Whilst going beyond the scope of our paper, understanding why weight decay leads to uniform rotation rates is definitely an interesting question to investigate in the future. We believe it will require a precise study of the equilibrium point to which the norm of each layer’s weights converges (i.e. the point where weight decay and the gradients cancel each other, such that the norm of a layer’s weights remains constant). In a sense, even if we are still unable to answer this question, we appreciate that our work raised such question in reviewer’s mind. Indeed, this question reveals that our work, by pointing the tight connection between rotation rates and generalization, opens several novel research avenues that could end up in disclosing key ingredients and methods regarding the control of network generalization.
>
> R8) We hope that the clarification of our analysis of adaptive gradient methods (cfr. comment to all reviewers) makes our methodology clearer. The reason why the five different adaptive gradient methods are applied on 5 different tasks, is because we do not compare the different adaptive methods to each other, but we rather compare each adaptive gradient method to SGD, which has been applied on the 5 tasks (cfr. Figure 4).
>
>
>
> We hope that this rather long rebuttal will convince the reviewer of the significance of our work. We of course are open to further discussion. We thank the reviewer again for his extensive review of our paper.

---

> > ### Comment · AnonReviewer2 · 2018-11-28
> > **Thank you for your response**
> >
> > Thank you for the response and clarifications. I intend to provide a more detailed response once I have time to review the changes. However, in the interest of time, I wanted to follow up with a more immediate question.
> >
> > With regards to point 4, on the technique used to measure rotation rates and clarifications on manifold optimization, I do not feel that this has been addressed in your response. Could you please include here which part of your comment you are referring to?
> >
> > In short, I believe I can reduce my issues with section 3.1 to the following:
> >
> > - You point out that each layer optimizes a global loss function making layer-level training speed ill-posed. How exactly does layer rotation resolve this?
> > - Why is flattening the weight vector a reasonable way to measure this? By ignoring matrix structure we throw away permutation invariances (see initial review comment) and other properties.
> > - Why is it reasonable to model the optimization as taking place on a manifold without matrix structure?
> >
> > For the first point, I acknowledge that in your response you seem to state that you are not suggesting that layer rotation rates is a good measure of layer training speed? But I think that the paper does not represent this. For example, Section 3.1, which introduces layer rotations, is titled "How can we measure layer-level training speed?".
> >
> > I will follow up with additional comments after I have looked through the revisions and comments in greater detail.

---

> > > ### Author Response · Authors · 2018-12-02
> > > **Citations from our initial rebuttal**
> > >
> > > [1] “Moreover, we absolutely agree that correctly measuring layer-level training speed remains an open problem that we do not solve in this paper.”
> > > [2] From the abstract: “... layer rotation rate, a tentative measure of layer-level training speed,...”
> > > From the introduction: “Our study starts from an educated guess about how to measure layer-level training speed appropriately,...”
> > > From the introduction of Section 3, where layer rotation rates are introduced: “our work starts by making an educated guess about how to measure layer-level training speed...”
> > > Using "How can we measure layer-level training speed?" as a title was done in order to highlight this pertinent but neglected question. Indeed, while this question is fundamental e.g. for works studying vanishing and exploding gradients (which are all about how fast all layers train), we are not aware of any paper that actually asks this question explicitly. If the reviewer believes this title is misleading and could lead the reader to think that layer rotation rates are the correct measure of layer-level training speed, we can of course change it in the final version, as this would not affect the paper’s contributions.
> > > [3] “Indeed, because layer rotation rates have such a consistent impact on model generalization and speed of convergence, they probably are related to a more fundamental aspect of learning, and to layer-level training speed in particular.”
> > > [4] “Hence, the useful guidelines emerging from our work to tune meta-parameters, and the fundamental insights we provide around weight decay and adaptive gradient methods remain valuable even if our intuitions are wrong, i.e. if layer rotation rates have no relation at all with layer-level training speed.”

---

> > > > ### Comment · AnonReviewer2 · 2018-12-09
> > > > **An updated response**
> > > >
> > > > Thank you for the detailed and clarifying response. I have taken some time to reread the paper in light of these comments and hope to provide an adequate response here.
> > > >
> > > > First, thank you for your comments with regards to flattening the weight vector/manifold optimization. I think that we are in agreement on most important points.
> > > >
> > > > With regards to the presentation of layer rotation as a solution to layer training speed. I must respectfully disagree with your position. I still believe that the paper may easily mislead a reader into believing that you are aiming (and even claiming) to address this problem exactly. Some examples (some of which you raised):
> > > >
> > > > - The paper title: "An experimental study of layer-level training speed [...]"
> > > > - From introduction: "Our work thus also contributes to the open problem of correctly measuring layer-level training speed"
> > > > -From introduction: "While the influence of layer-level training speed [...] our observations suggest that its impact [...]" *
> > > > - In related: "These works, however, do not use weight rotation as a measure of layer-level training speed"
> > > >
> > > > Perhaps this is largely an issue of consistency and interpretation, but I think it is one that based on reviewer responses and your comments should be resolved. I am not convinced (though it seems plausible) that layer rotation rates can be directly linked to layer training speed in general but in my opinion the paper puts significant weight behind this claim.
> > > >
> > > > I also understand the stance you mean to take: "Actually, the proposition that layer rotation rates could be a measure of layer-level training speed should not be considered as an argument supporting our three claims, but rather as a probable consequence of these claims". As I see it, this seems to assume that layer-level training speed would be linked to generalization but I do not believe you provide any direct evidence for this (in citation or otherwise). See * above.
> > > >
> > > > I would like to emphasize that I think this paper presents some interesting empirical findings. I think that several parts could be expanded in interesting ways (which speaks in-part to the quality of the paper). I think one important direction would be investigating recent improvements to adaptive methods (such as decoupling weight decay) to see if layer rotation can provide some insight here. On a related note, I would still like to see some formal explanation for the link between weight decay and layer rotation speed.
> > > >
> > > > I have given careful consideration to this review and have ultimately decided to retain my original score. This paper presents interesting observations and I hope that some relatively small changes could greatly improve things.

---

> > > ### Author Response · Authors · 2018-12-02
> > > **Further discussion of point 4 raised by the reviewer**
> > >
> > > Thank you for your fast reaction.
> > > We see two parts in your 4th point. The first part questions the relevance of layer rotation rates: “It is not obvious to me that layer rotation speed is necessarily a good measure of training speed”, while in the rest of the comment you mention intuitions and suggestions that could potentially improve the measure and the optimization algorithm.
> > >
> > > In our reply to all reviewers, only the first part has been explicitly addressed. To your request, we restate, with pointers towards our initial reply, how we address this first question, related to the relevance of layer rotation rates. In our reply, we first clarify that we do not claim that layer rotation rates are the correct measure of layer-level training speed, but rather an educated guess [1] (we believe this is also made clear in the current version of our paper, cfr [2]). We then highlight a solid inductive argument in favor of layer rotation rates being a pertinent (i.e. not too far from the true) measure of layer-level training speed: its consistent impact on fundamental aspects of training (e.g. generalization and the height of plateaus in training curves) [3]. Going further, we argue that the fact that layer rotation rate might not be the correct measure of layer-level training speed does not invalidate our claims [4].
> > > Hence, the conclusion of our initial rebuttal is that, whilst being a nice inductive outcome from our experiments, layer rotation rate being a good metric of layer-level training speed (which we understood was the main issue raised by the reviewer) is not required to validate our claims.
> > >
> > > Regarding the propositions made in the second part of your 4th point, we think they are interesting directions for future work rather than essential modifications to implement in our current paper.
> > >
> > > First, we want to highlight that the flattening of the weight vector has been motivated by the will to (i) define a simple scalar rotation-based metric that reflects how all the weights of a *specific* layer change along the optimization, and (ii) offer a straightforward way to control the layer updates during training, to enable a systematic study of its relation to generalization. While more rigorous metrics for studying layer-level dynamics exist (e.g. SVCCA at NIPS2017), they usually fail to satisfy these two criteria.
> > >
> > > We now expose some reflexions about the specific and constructive/relevant propositions of the reviewer.
> > >
> > > Proposition 1: adding invariances to the metric that make use of the matrix structure, in particular invariance to permutation.
> > > - It is important to note that even if it makes sense for our metric to be invariant to permutation, it is not necessarily needed to implement this invariance explicitly. Indeed, if permutations do not occur in the data we study, incorporating invariance to permutation would not add anything to the study. The occurrence of permutations during stochastic gradient descent optimization seems quite improbable to us.
> > > - To further comment on the invariance highlighted by the reviewer, we point out that a network is not invariant to permutations in a single layer but only to permutations in two adjacent layers that counteract each other.
> > >
> > > Proposition 2: Optimizing on the Stiefel manifold instead of a spherical manifold (at the layer level).
> > > - Optimizing on the manifold of orthonormal matrices (e.g. with projection based on https://hal.archives-ouvertes.fr/hal-00651608/PDF/absil-malick.pdf ) would be a very interesting path for further investigation. Especially because of its tight relations with previous works on dynamical isometry and orthogonal initializations.
> > > - We are however not sure to see how such alternative optimization procedure would improve the study of layer-level training speed. In particular, what metric of layer-level training speed would be used and controlled during the optimization procedure? While less critical, it is also interesting to note that there does not exist an analytical formula to compute distances on the Stiefel manifold.

---

> ### Author Response · Authors · 2018-11-28
> **Detailed answer to the comments of the reviewer (1/2)**
>
> Thank you very much for your interest in our paper and the pertinent questions that you raised. The bulk of our rebuttal is contained in the two comments addressed to all reviewers. We add here answers to your specific questions.
>
> R1) Indeed, the experiment of Figure 1 is independent of our work around layer rotation rates. We see it as a motivating example that shows that studying layer-level training’s impact on generalization is indeed pertinent and deserves our attention. Thanks to the experiment’s simplicity (only one layer is trained at a time), the complex problem of defining an appropriate measure of layer-level training speed is avoided. We’ve rewritten our analysis of Figure 1 to make our reasoning clearer. In particular, the conclusion of the experiment has been rephrased to “The generalization ability induced by a layer’s training is thus heavily affected by how the other layers transform the input and feedback signals it receives” (cfr. caption of Figure 1).
>
> R2) Thanks for your interest in our experiment. As suggested by the reviewer, we’ve ran the experiment of Figure 1 with orthogonal initial weights. The results are presented in supplementary material, Section A.1.1. Whilst slightly different from the plot obtained with non-orthogonal initialization, the message conveyed by the figure remains unchanged. The reviewer could be interested in our complementary analysis presented in Section A.1.2., that provides insights as to why the first layers generalize better than the last layers in this experiment.
>
> R3) Thanks for the reference, we’ve added it to the related work section. The main difference with our work lies in the metric used for measuring layer-level training speed. Indeed, these works focus on improving how signals propagate through the network (during forward and backward pass), and study the gradient signal specifically. Layer rotation rates, which are our subject of study, do not only depend on the gradient signal, but also on the layer’s weight vector.
>
> R4) This comment is answered in our general comment addressed to all reviewers.
>
> R6) We are not sure to understand what is meant by ‘each weight matrix has the same effective learning rate’, as this doesn’t seem to be in accordance with the definition of effective learning rate provided in [4][5][6]. We however suspect that the reviewer refers to some kind of update normalization. We’ve added results in Section 6 showing that methods based on update normalization (such as the one proposed by the reviewer) do not have a consistent impact on generalization. We argue that this consistency is what makes layer rotation rates unique.

---

### Official Review · AnonReviewer3 · 2018-11-02
**Nice empirical study with a layerwise perspective on training/generalization**

**Rating:** 6
**Confidence:** 3

**Review:**

Pros:
Overall, this is a nice empirical paper with a reasonably extensive set of experiments. It is interesting that, among networks that train to ~100% with Layca, the best generalizing ones tend to have balanced training between layers (Fig. 2), and that tuned SGD does not generalize as well as Layca (Fig. 4). I think this paper’s focus on discrepancies in training & generalization originating from layers of a deep network is an interesting and important topic of study that warrants further empirical and theoretical investigation from the community. I think the work already has some interesting results and will encourage further investigation.

Cons:
--Would appreciate greater discussion of the originality of the results; in particular, a more upfront discussion (which is currently concisely presented in the supplementary) regarding algorithms that are similar to Layca when less crucial steps are dropped, e.g. Yu et al 2017 and Ginsburg et al 2018.
--After reading the paper, I don’t feel especially convinced that rotation (of the flattened weight matrix) is the best quantity to analyze training dynamics of a single layer. Could there be greater discussion & motivation for this, and in particular, relationship to work where weights are parameterized using orthogonal matrices, or even orthogonal initialization?

Some minor comments:
--Would have appreciated a discussion of the learning rate schedule (as well as other experimental details, e.g. loss function used and what role this plays) and whether networks with lower learning rates would need to be trained longer.
--Greater discussion of why the first and last layer(s) do not experience the same rotation rate as other layers and if there would be better generalization if they did.

---

> ### Author Response · Authors · 2018-11-28
> **Detailed answer to the comments of the reviewer**
>
> Thank you very much for your encouraging comments, we really appreciate it. The bulk of our rebuttal is contained in the two comments addressed to all reviewers. We add here answers to your specific questions.
>
> Replies to ‘Cons’:
> -- Algorithms similar to Layca are now mentioned in the related work section, with explicit mention that they may even be equivalent in some practical applications. Note that the similarity between Layca and previous works does not really affect the originality of our contribution since the novelty of our work lies in the claims derived from the extensive analysis performed based on Layca, and not in the design of Layca itself.
> -- The pertinence of measuring layer-level training speed through the rotation rate of weight vectors is discussed in depth in the comment addressed to all reviewers. Using orthogonal weights in deep networks was proposed to deal with the vanishing/exploding gradients problem, because such weights better preserve the norm of signals through multiplication. This assumes implicitly that only the norm of input and gradient signals matter to measure layer-level training speed, and not the norm of each layer’s weights. The section (Section 6) we’ve added to the paper shows that such assumption is probably wrong. Indeed, metrics of layer-level training speed that do not depend on the norm of the weights do not demonstrate a consistent relation to generalization.
>
> Replies to ‘Some minor comments’:
> -- The learning rate schedules are now described in the Supplementary Material. The approach is very standard: reduce the learning rate by a factor of 5 or 10 at specific epochs during training (e.g. [1]). We don’t believe it is necessary to train networks with lower learning rates for a longer period. First, as shown in Figure 3, lower learning rates sometimes required shorter training time (which is quite unintuitive), because the network doesn’t get stuck in high loss curve plateaus in such cases. Second, as detailed in Section B.4, the networks trained with lower learning rates, and for which layer-wise angle deviation curves are presented in the paper, all achieve ~100% training accuracy.
> -- This is an interesting research question, that we hope to elucidate in future work.
>
> We hope our rebuttal answers the reviewer’s doubts, and are open to further discussion. Thank you again for your efforts and positivity.
>
> [1] Deep Residual Learning for Image Recognition; K. He et al., CVPR 2016

---

### Author Response · Authors · 2018-11-28
**Comment for all reviewers (1/2): a change in perspective that clarifies the role of layer rotation rates in our paper**

The overall positivity expressed in the three reviewers' feedback has been much appreciated, and we thank them for that. We also took very seriously the reviewers’ reluctance to turn this optimism into high evaluation scores. After careful analysis, we believe this reluctance is rooted in a few presentation issues, resulting in a fundamental change in the reader’s perspective. In this rebuttal, we first clarify this issue, before providing a more detailed answer to each reviewer’s specific requests.

Our paper defends three claims:
i) layer rotation rates (a quantity defined in the paper to monitor how weight vectors change along the training, whatever the optimization strategy) have a consistent and substantial impact on generalization;
ii) weight decay is a key ingredient for enabling the emergence of beneficial layer rotation rates during SGD training; and
iii) adaptive gradient methods' impact on generalization and training speed does not result from parameter-level adaptation, but rather from layer-level adaptation of the learning rate.

In the initial version of the paper, the analysis and experiences leading to these claims were introduced and motivated by presenting layer rotation rates as a tentative measure of a more fundamental aspect of learning: layer-level training speed (as discussed in Section 3.1). We perceive from the reviewers’ feedback that this formulation has been a source of disagreement (legitimately, the reviewers are not convinced a priori that the layer rotation rate measures the layer-level training speed), but also a source of confusion with respect to the methodology of the paper: unfortunately, our writing could suggest that the relation between layer rotation rates and layer-level training speed was essential for the contributions of the paper to be of any value.

Actually, the proposition that layer rotation rates could be a measure of layer-level training speed should not be considered as an argument supporting our three claims, but rather as a probable consequence of these claims. To clarify this perspective shift, we think it is useful to reconsider the fundamentals of the scientific method itself. Scientific progress is based on two forms of reasoning: deductive and inductive reasoning. Deductive reasoning consists in a rational argument that proves a claim by deducing it from a known theory. Inductive reasoning consists in making a claim more probable by providing empirical evidence in favor of it. Inductive reasoning is a fundamental part of science, which lead for example to Newton's law of gravity. This law contributed to science like only a few others did. Despite this, Newton was not able to explain why such a law would exist in the universe. He was only convinced that it did because all his observations were in accordance with it.

In the context of an infinitely more modest endeavor, our paper really shares Newton’s spirit. For example, we claim that layer rotation rates have a consistent impact on generalization and model convergence. Even though we cannot provide a convincing explanation of why this is the case, this claim still has scientific value: all our experiments are in accordance with it (~250 different training routines have been performed to construct our figures). Hence, the useful guidelines emerging from our work to tune meta-parameters, and the fundamental insights we provide around weight decay and adaptive gradient methods remain valuable even if our intuitions are wrong, i.e. if layer rotation rates have no relation at all with layer-level training speed.

Moreover, we absolutely agree that correctly measuring layer-level training speed remains an open problem that we do not solve in this paper. However, we believe our paper still contributes to this research question. Indeed, because layer rotation rates have such a consistent impact on model generalization and speed of convergence, they probably are related to a more fundamental aspect of learning, and to layer-level training speed in particular. We are not aware of any other metric of layer-level training speed that demonstrated such consistency. To make this observation more explicit in the revised version, Section 6 has been added to show that monitoring the norm of each layer’s weight updates, another common metric, does not demonstrate such consistency at all.

We are happy that the reviewing process highlighted the presentation issues of our paper, and thank the reviewers once again. We’ve put lots of efforts to present our claims and methodology more clearly in the revised version of the paper. We hope that the reviewers will agree that this perspective shift indeed answers their main criticisms, and hence will change their evaluation of our paper.

---

### Author Response · Authors · 2018-11-28
**Comment for all reviewers (2/2): highlighting the contribution provided by our analysis of adaptive gradient methods**

A second common trend in the reviewers’ feedback is the absence of comments (positive or negative) regarding our analysis of adaptive gradient methods (Section 5.2, or claim (iii)). This surprised us, as our experiments provide convincing evidence that the common understanding of adaptive gradient methods’ inner workings is misleading and misses their key role in deep learning applications.

Indeed, adaptive gradient methods were introduced with a focus on parameter-level adaptation of the learning rate in order to improve optimization in the presence of sparse gradients or optimization in online and non-stationary settings [1]. Our work offers and empirically validates a radically different understanding of adaptive gradient methods' role in deep learning, that solely relies on their impact on the rate at which each layer's weights change during training. This new perspective reveals that adaptive gradient methods' parameter-level adaptation of the learning rate is not necessary (layer-level adaptation suffices), and that their impact on DNN training will probably be better explained by further studying the layer-level training speed rather than the sparsity of gradients or the online aspect of the optimization procedure.

We believe the experiment presented in Section 5.2 deserves to be presented to the community, as it will be valuable to many works around adaptive gradient methods, a very active research topic (Adam [1] has been cited more than 14.000 times in the last 4 years, and many open questions remain to be answered e.g. [2][3][4][5][6][7]). Sadly, the initial version of our paper did not make this contribution explicit enough (which probably explains why the reviewers did not comment on it). The revised version of our paper should be explicit in that regard (see paragraph 3 Section 5.2).

[1] Adam: A method for stochastic optimization; D. Kingma and J. Ba, ICLR 2015
[2] The Marginal Value of Adaptive Gradient Methods in Machine Learning; C. Wilson et al., NIPS 2017
[3] On the Convergence of Adam and Beyond; S. Reddi et al., ICLR 2018 https://openreview.net/forum?id=ryQu7f-RZ
[4] Padam: Closing the Generalization Gap of Adaptive Gradient Methods in Training Deep Neural Networks; Anonymous, https://openreview.net/forum?id=BJll6o09tm
[5] A unified theory of adaptive stochastic gradient descent as Bayesian filtering; Anonymous https://openreview.net/forum?id=BygREjC9YQ
[6] AdaShift: Decorrelation and Convergence of Adaptive Learning Rate Methods; Anonymous https://openreview.net/forum?id=HkgTkhRcKQ
[7] Adaptive Gradient Methods with Dynamic Bound of Learning Rate; Anonymous https://openreview.net/forum?id=Bkg3g2R9FX

---

### Meta-Review · Area_Chair1 · 2018-12-14
**An interesting question but there are issues with the presentation and analysis**

**Confidence:** 3
**Recommendation:** Reject

**Metareview:**

Dear authors,

The reviewers all appreciated the question you are asking and the study of the impact of each layer is definitely an interesting one.

They were however uncertain about the actual metrics you used to emphasize your points. Further, as you noted, there were quite a few presentation issues that led to skepticism of the reviewers, despite them spending quite a bit of time reading the paper and engaging in discussion.

Hence, I regret to inform you that your work is not yet ready for publication. A more focused analysis would be a great addition to the questions you raise.